**METHOD**

# iDNA-ABF: multi-scale deep biological language learning model for the interpretable prediction of DNA methylations

Junru Jin[1,2], Yingying Yu[1,2], Ruheng Wang[1,2], Xin Zeng[3,4], Chao Pang[1,2], Yi Jiang[1,2], Zhongshen Li[1,2], Yutong Dai[3,4], Ran Su[5], Quan Zou[6], Kenta Nakai[3,4*] and Leyi Wei[1,2*]

*Correspondence:
knakai@ims.u-tokyo.ac.jp;
weileyi@sdu.edu.cn

[1] School of Software, Shandong University, Jinan 250101, China
[2] Joint SDU-NTU Centre for Artificial Intelligence Research (C-FAIR), Shandong University, Jinan 250101, China
[3] Human Genome Center, The Institute of Medical Science, The University of Tokyo, Tokyo 108-8639, Japan
[4] Department of Computational Biology and Medical Sciences, The University of Tokyo, Kashiwa 277-8563, Japan
[5] College of Intelligence and Computing, Tianjin University, Tianjin 300350, China
[6] Institute of Fundamental and Frontier Sciences, University of Electronic Science and Technology of China, Chengdu 610054, China

## Abstract

In this study, we propose iDNA-ABF, a multi-scale deep biological language learning model that enables the interpretable prediction of DNA methylations based on genomic sequences only. Benchmarking comparisons show that our iDNA-ABF outperforms state-of-the-art methods for different methylation predictions. Importantly, we show the power of deep language learning in capturing both sequential and functional semantics information from background genomes. Moreover, by integrating the interpretable analysis mechanism, we well explain what the model learns, helping us build the mapping from the discovery of important sequential determinants to the in-depth analysis of their biological functions.

**Keywords:** DNA methylation, Deep learning, Multi-scale information processing, Interpretable analysis

## Background

DNA methylation is significant for the development and plays an important role in gene silencing, protection against spurious repetitive element activity, genomic stability during mitosis, and parent-of-origin imprinting [1]. Moreover, alteration of the DNA methylation pattern caused by the environment and aging may contribute to the development of disease, especially cancer [2, 3]. Currently, 5-methylcytosine (5mC), N6-methyladenosine (6mA), and 4-methylcytosine (4mC) are three main DNA methylation types, named according to the type of nucleotide, the type of molecule added, and the position of modification within the nucleotide [4]. Different methylations have diverse functional mechanisms. For example, among them, 5mC is generated by binding methyl groups at the fifth site of cytosine (C). It is associated with transcriptional inhibition, and thus with classical epigenetic phenomena such as genomic imprinting and X chromosome inactivation [5]. 6mA, usually with methylation at the sixth position in adenosine (A), plays a

crucial role in chromosome replication, cell defense, cell-cycle regulation, and transcription [6]. It has been extensively detected in viruses, bacteria, protists, fungi, algae, etc. As the other important epigenetic modification, 4mC protects host DNA from the degradation of restriction enzymes and corrects prokaryotic DNA replication errors, and controls the DNA replication and cell cycle of prokaryotes [7]. Therefore, DNA methylation identification is fundamentally essential for revealing the functional mechanisms.

DNA methylation can be determined experimentally through next-generation sequencing (NGS) approaches such as whole-genome bisulfite sequencing (WGBS) [8] or reduced-representation bisulfite sequencing (RRBS) [9]. The techniques can determine the global genomic distribution of DNA methylations at the nucleotide level and provide golden standard datasets for DNA methylation-related downstream task analysis. However, the detection of DNA methylation using traditional experimental techniques is often costly and time-consuming [10]. In addition, bisulfite sequencing cannot profile DNA methylation in repetitive genomic areas due to short-read sequencing [11, 12]. Thus, recent research is more focused on developing computational approaches, particularly machine learning-based approaches, to detect DNA methylations directly using genomic sequences. These methods formulate DNA methylation identification as a binary prediction task and train machine learning models to distinguish true methylation sites from non-methylation sites.

Over the last few decades, a series of sequence-based approaches using either traditional machine learning or deep learning are well developed for the prediction of DNA methylations. Taking 4mC methylation prediction as an example, Tang et al. proposed DNA4mC-LIP, an ensemble learning method by combining six existing predictors through a linear integration strategy to make predictions [13]. DeepTorrent [14] is a deep learning-based predictor that integrates inception module, attention module, and transfer learning to improve the predictive performance of 4mC sites. As the prediction of 6mA sites, MM-6mAPred [15] makes use of the transition probability between adjacent nucleotides based on a Markov model. To simplify the model construction, SNNRice6mA [16] builds a simple and lightweight deep learning model using Convolutional Neural Network (CNN) to identify 6mA sites in the rice genome. Later on, Li et al. proposed Deep6mA [17], a hybrid deep learning network of CNN and Long Short-Term Memory (LSTM), with more accurate 6mA prediction. BERT6mA [18] is a similar model but uses transformer to build predictive models, demonstrating the effectiveness of natural language processing techniques with applications in 6mA prediction. As for 5mC site detection, iPromoter-5mC fuses the results of several models that predict the one-hot encoded sequence through full connection layers [19]. BiLSTM-5mC mainly uses Bidirectional Long Short-Term Memory (BiLSTM) to extract features of sequences encoded by nucleotide property and frequency for the 5mC prediction [20]. However, most existing approaches can only distinguish one single type of DNA methylation. They are difficult to generalize to other methylation types. iDNA-MS [21] is the first machine learning predictor, which is designed for generic detection of different methylations across different species. The iDNA-MS utilizes manual features such as K-tuple nucleotide frequency component and mono-nucleotide binary encoding with traditional machine learning algorithms like support vector machine (SVM) and random forest (RF). The shortcoming of iDNA-MS is that the feature design highly requires a lot of

prior knowledge and meanwhile lacks adaptability among different methylation prediction tasks. To address this problem, in our previous work, we designed a deep learning model, namely iDNA-ABT [22] that uses the architecture of Bidirectional Encoder Representations from Transformers [23] (BERT) to automatically and adaptively learn distinguishable features and make relatively accurate predictions for different methylation types in different species.

As seen above, more and more research efforts attempt to explore the potential of deep learning in the prediction of DNA methylations, and certain progress has been made in the improvement of predictive performance. However, existing deep learning predictors have not fully explored the power of feature representation learning, especially in the discovery of key sequential patterns that are important for elucidating the DNA methylation mechanisms. This also results in the deep learning models with poor interpretation and not being able to dig out the important influence of sequence-based models in DNA methylation prediction. On the other hand, existing approaches fail to answer other important questions: (1) whether background genomic sequences contain extra distinguishable information that can guide the development of DNA methylations, and (2) whether DNA methylation occurring exists the conservation and specificity of sequential patterns across species or cell lines from computational perspectives is also a key problem.

With the development of natural language processing, there are some advanced techniques such as BERT [24] that are capable of sufficiently exploring and learning high-latent contextual information in natural language texts. Inspired by this, we here consider genomic sequences as "biological texts" and take different-scale sequential determinants as different "biological words". Therefore, we propose iDNA-ABF, a multi-scale biological language learning model to successfully build the mapping from natural language to biological language, and the mapping from methylation-related sequential determinants to their functions. Specifically, we introduce a model well pretrained with large-scale genomic sequences to learn biological contextual semantics and propose a multi-scale processing strategy to capture discriminative methylation information from different scales. We further utilize adversarial training and transfer learning to improve the predictive performance and enhance the robustness of our model. Benchmarking results on seventeen datasets across different methylations and species show that our model significantly outperforms the state-of-the-art sequence-based methods. Importantly, our model provides interpretable prediction and analysis at sequence level by exploring the local sequential characteristics based on attention mechanisms. The results reveal that our model can accurately and adaptively locate the sequential regions that are closely associated with methylations, demonstrating that there might exist "biological language grammars" that are participating in functional regulations in cellular progress.

## Results

### The proposed iDNA-ABF outperforms the state-of-the-art methods

To evaluate the performance of our proposed iDNA-ABF, we compared it with four state-of-the-art predictors, including iDNA-ABT, iDNA-MS, BERT6mA, and Deep6mA. Of the four predictors, the former two (iDNA-ABT and iDNA-MS) are generic predictors for different methylation predictions while the other two (BERT6mA and Deep6mA)

are originally designed for 6mA site prediction. The reason to include the two 6mA predictors for performance comparison is that they are the state-of-the-art predictors based on deep learning. Moreover, their models are flexible and can be well extended for other methylation predictions like 5hmC and 4mC, not only for 6mA. All the compared predictors were respectively trained on seventeen training datasets across different species and different methylation types, and evaluated on the corresponding independent testing datasets (see "Datasets" section for details). The evaluation results in terms of ACC and MCC are shown in Fig. 1A and B, respectively. The detailed results in other metrics such as SN and SP are presented in Additional file (Additional file 1: Table S1). As clearly seen in Fig. 1A and B, our model outperforms the four existing predictors on 15 out of 17 datasets (Additional file 1: Table S1), with only two exceptions—*5hmC_M.musculus* and *6mA_A.thaliana*, in which our model is actually comparable with the best predictors as well. To be specific, the average ACC of our model on all datasets is higher than that of two runner-up predictors iDNA-ABT by 1.34% and BERT6mA by 3.73%, respectively. In particular, on the three datasets (*4mC_C.equisetifolia*, *4mC_S.cerevisiae*, and *6mA_S. cerevisiae*), our iDNA-ABF performs better than the existing predictors with a relatively large margin, leading by 3.28–14.75%, 1.88–3.59%, and 1.48–4.23% in ACC, respectively. Similar results are observed in terms of MCC. To this end, the results demonstrate that our iDNA-ABF is superior to the state-of-the-art approaches for the generic prediction of DNA methylations. More importantly, it shows robust performance across species under the three methylation types.

To validate the robustness of our model, we further illustrated the ROC and PR curves of the predictors on four datasets (*4mC_C.equisetifolia, 5hmC_M.musculus, 6mA_C. equisetifolia*, and *6mA_F.vesca*) as presented in Fig. 1C–F, respectively. We can see that our iDNA-ABF has the highest AUC and AP in all four datasets. Specifically, the average AUC and AP values of our model on the four datasets increase by about 1.39–2.81% and 0.1–13.8% as compared to the other predictors, respectively. The results further demonstrate the robust performance of our model in DNA methylation prediction tasks. The ROC and PR curves on the other datasets can be found in Additional file (Additional file 1: Fig. S1 and Fig. S2). To intuitively discuss why our iDNA-ABF performs better than the other approaches, we further visualized the distribution of feature representation space of our iDNA-ABF and the second-best predictor iDNA-ABT on the above four datasets (*4mC_C.equisetifolia, 5hmC_M.musculus, 6mA_C.equisetifolia*, and *6mA_F.vesca*) using Uniform Manifold Approximation and Projection (UMAP) [25], a widely used visualization tool that reveals the essential data characteristics through dimensionality reduction. Note that the UMAP visualization results on the other datasets can be found in Additional file (Additional file 1: Fig. S3). Figure 1G and H illustrate the feature space distribution of our iDNA-ABF and iDNA-ABT, respectively, in which each point represents each sample; methylation sites (positive samples) are annotated with red color while non-methylation sites (negative samples) with blue color. As seen from Fig. 1G, our model separates the positive and negative samples clearly and every class clusters together rather than disperse, while in Fig. 1H, the positive and negative samples in the feature space of the iDNA-ABT are distributed almost connected, which is not easy to circle the boundary for each class. By comparing Fig. 1G and H, we found that the two classes are distributed more clearly in the feature space of our iDNA-ABF

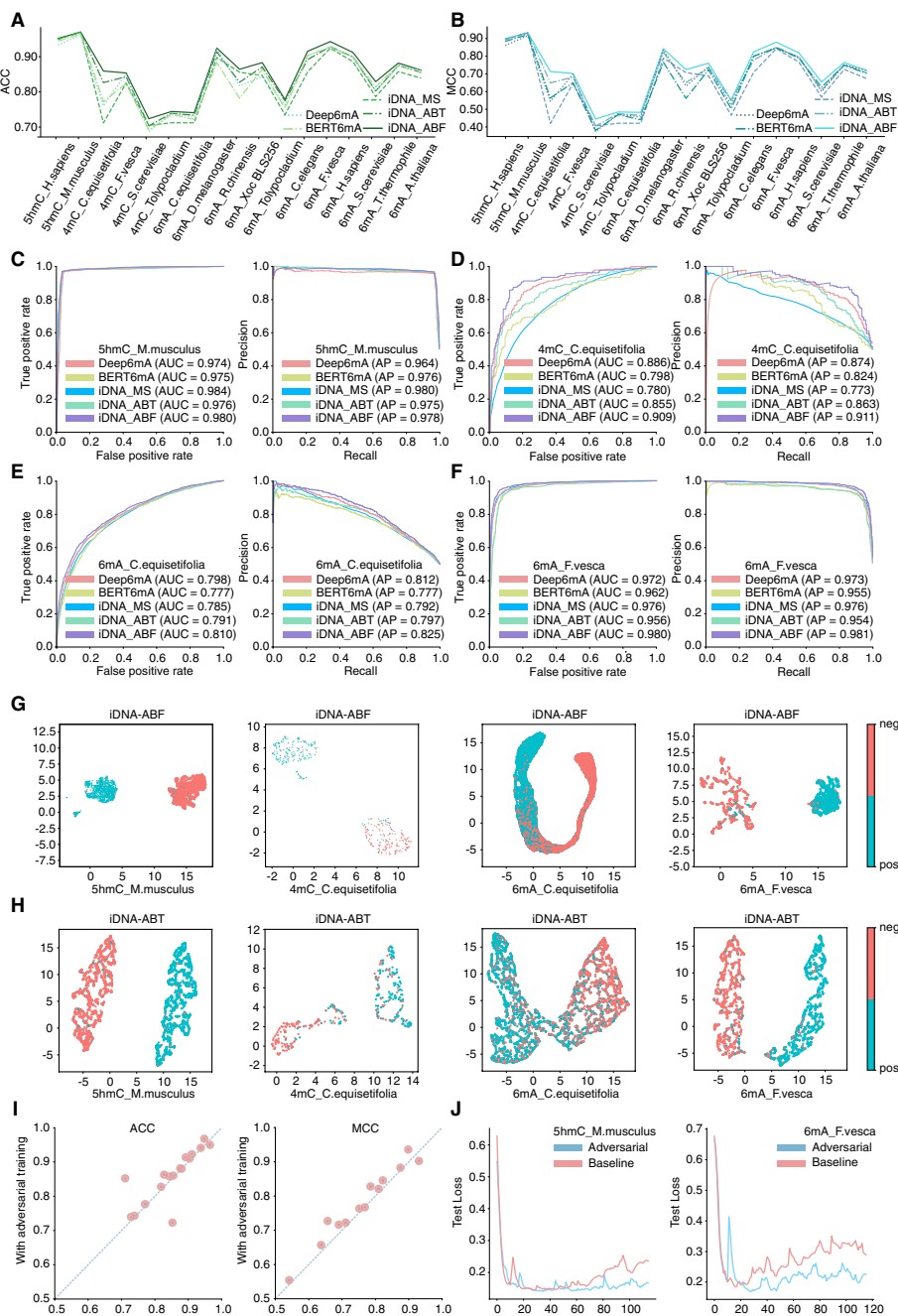

**Fig. 1** Performance comparison between iDNA-ABF and other existing methods. **A** and **B** represent the ACC and MCC values of our proposed iDNA-ABF and other existing methods including iDNA-ABT, iDNA-MS, BERT6mA, and Deep6mA on 17 benchmark independent datasets, respectively. **C** The ROC and PR curves of our proposed iDNA-ABF and other existing methods in *5hmC_M.musculus*. **D** The ROC and PR curves of our proposed method and other existing methods in *4mC_C.equisetifolia*. **E** The ROC and PR curves of our proposed iDNA-ABF and other existing methods in *6mA_C.equisetifolia*. **F** The ROC and PR curves of our iDNA-ABF and other existing methods in *6mA_F.vesca*. **G** and **H** represent the feature space distribution (with UMAP visualization) of iDNA-ABF and iDNA-ABT in *5hmC_M.musculus, 4mC_C.equisetifolia, 6mA_C. equisetifolia*, and *6mA_F.vesca*, respectively. Negative (in red color) and positive (in blue color) represent non-methylation and true methylation samples, respectively. **I** The MCCs and ACCs of the models with and without adversarial training on 17 benchmark independent datasets, respectively; each point in the figure represents each dataset. **J** Learning curves of the model with and without the use of adversarial training on *5hmC_M.musculus,* and *6mA_F.vesca*

as compared to the state-of-the-art iDNA-ABT. This demonstrates that our model learns better feature representations from different class samples, possibly due to the well pre-trained model in our model construction, helping us capture more high-latent contextual semantics information from millions of background genomic sequences.

**Adversarial training enhances the predictive performance and the robustness of iDNA-ABF**

Adversarial training is an important component of our iDNA-ABF. To investigate the effectiveness of the adversarial training, we compared our original iDNA-ABF with the model without the use of adversarial training. The results of the 17 independent datasets are illustrated in Fig. 1I where each dot represents each dataset. As seen, our original iDNA-ABF (with adversarial training) generally achieves better performance than that without adversarial training. To be specific, by introducing adversarial training, the performance improvement in ACC and MCC can be observed on 14 out of 17 datasets, and 15 out of 17 datasets, respectively. This indicates that adversarial training can enhance prediction performance. The results on other metrics (SN, SP, and AUC) can be found in Additional file (Additional file 1: Fig. S4). What is more, to intuitively show the importance of adversarial training in model optimization, we further analyzed the learning curves during the training process. Figure 1J shows the curves of the models with and without adversarial training on two datasets (*5hmC_M.musculus* and *6mA_F.vesca*), randomly selected from the datasets. From Fig. 1J, we can see that the models with adversarial training achieve lower test loss than that without adversarial training although the loss reduction rate decreases more slowly than the models without adversarial training. Furthermore, using adversarial training the models maintain lower test loss in the later period of the training process while the models without adversarial training gradually begin to overfit, demonstrating that adversarial training enhances the robustness of our model in the DNA methylation prediction.

**Our iDNA-ABF reveals the methylation conservation across species at sequential level**

To investigate whether the methylated sequential patterns across different species are conserved or not, we firstly constructed the evolutionary tree for different species in the same methylation type using Lifemap [26]. As for 4mC methylation, Fig. 2A illustrates the evolutionary relationship of four species. It can be clearly seen that *Fragaria vesca* and *Casuarina equisetifolia* are evolutionary taxonomies, belonging to the common *Fabids*, while the other two species belong to *Saccharomyces*. An interesting observation is that our model exhibited similar performance in the species with evolutionary taxonomies. In *F. vesca* and *C. equisetifolia*, the ACCs of our model are 0.852 and 0.858, respectively; while in the other, their ACCs are 0.743 and 0.723. Next, we further analyzed the methylation sequential patterns of the four species using the probability-based motif visualization tool—kpLogo [27]. Figure 2B illustrates the sequential patterns in two evolutionarily close species (*F. vesca* and *C. equisetifolia*) while Fig. 2C shows that in the other two species. From Fig. 2B, we can see that the methylated sequential regions in the species are very similar, particularly enriched with CG content. From Fig. 2C, the similar results in the other two species can be observed. As for the 6mA methylation, we also found the similar conclusion with 4mC methylation (Additional file 1: Fig. S5). Overall, the results demonstrate that the methylated sequential patterns in species with

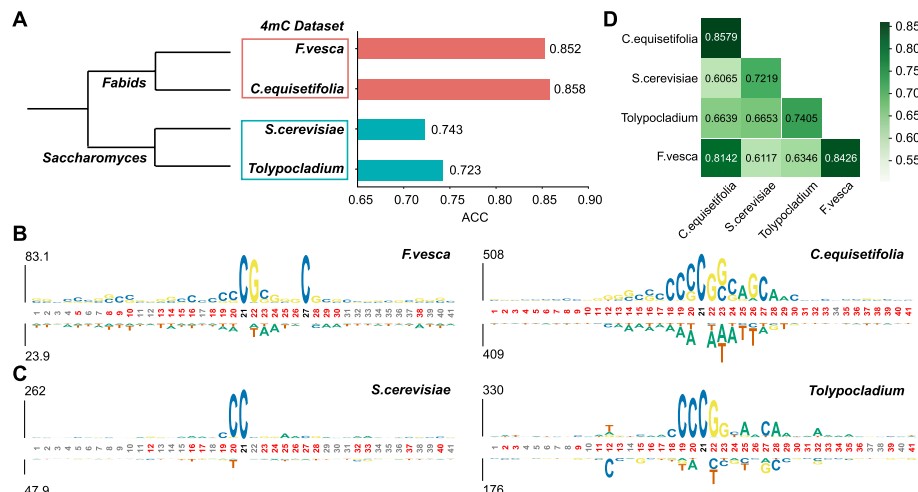

**Fig. 2** The relationship between methylation conservation and model accuracy across species. **A** Taxonomy tree and accuracy for four species in 4mC dataset. Two species (i.e., *F. vesca* and *C. equisetifolia*) with higher accuracy are grouped in red, while the other two species (i.e., *S. cerevisiae* and *Tolypocladium*) with lower accuracy are grouped in blue. **B** The motif logo analysis on *F. vesca* and *C. equisetifolia*. **C** The motif logo analysis on *S. cerevisiae* and *Tolypocladium*. **D** The accuracy heatmap of cross-species validation on 4mC dataset

evolutionary taxonomies might be conserved, thus contributing to the similar predictive performance; on the other hand, the methylation patterns in the species with far evolutionary relationship would be quite different.

Next, we further investigated the cross-species performance of our model to study their interrelationships between species; that is, we trained our model in one species and tested on the other. To avoid the problem of insufficient learning, we only trained our models on large datasets and tested on small datasets. The cross-species performances are illustrated in Fig. 2D, from which we can see that the performances within evolutionary taxonomies are significantly better than that without evolutionary taxonomies. The results further demonstrate that the methylation conservation at sequential level is positively correlated with evolutionary taxonomies.

## Multi-scale sequential design choice is more appropriate to elucidate methylation mechanisms

In our model, we proposed a multi-scale information processing strategy via using different *k*-mers to represent different "biological words" for feature representation learning. Therefore, we firstly validated how single-scale *k*-mers impact the predictive performance of our model. We compared different *k*-mers, ranging from 3-mer to 6-mer. The comparative results are illustrated in Fig. 3A, in which we can see that different *k*-mers indeed have their advantages on different datasets, respectively. There is no consistent result observed. It might be that the methylated sequential regions vary across species and methylation types in length. Therefore, using single-scale sequential patterns for feature representations cannot adaptively and sufficiently capture the inherent characteristics of methylations. To address this problem, we integrated different scales of *k*-mers as our model input, such as 3-mer + 6-mer, 4-mer + 6-mer, and 5-mer + 6-mer, and compared their performance as illustrated in Fig. 3B. It can be observed that the multi-scale

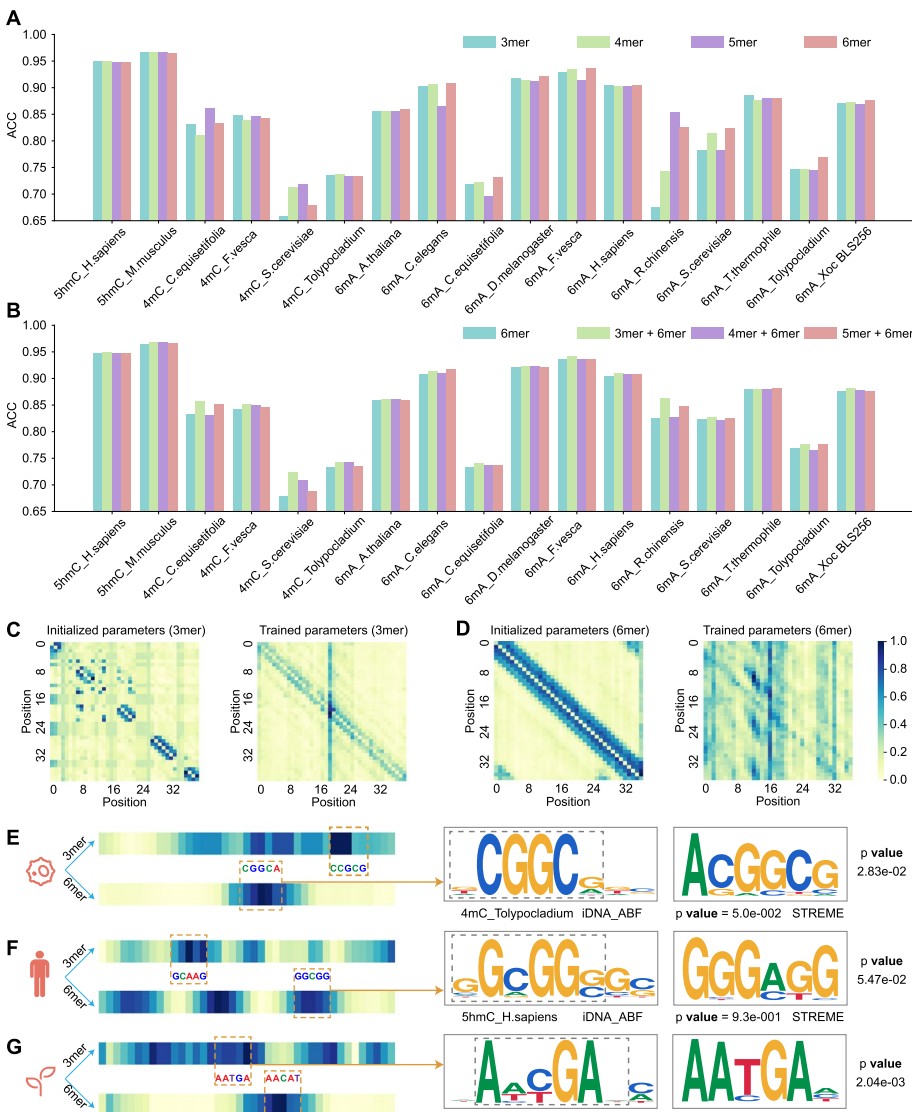

**Fig. 3** Interpretable analysis of multi-scale information processing. **A** The comparison of single scales including 3-mer, 4-mer, 5-mer, and 6-mer, respectively. **B** The comparison of multi-scale combinations. **C** The attention map to illustrate the information captured at 3-mer scale on one randomly selected sequence. Two sub-figures visualize the change of information captured before and after training, respectively. **D** The attention map to illustrate the information captured at 6-mer scale. **E–G** Interpretable illustrations of the motifs learnt by our model in three species covering three methylation types, including *4mC_Tolypocladium*, *5hmC_H.sapiens*, and *6mA_C.equisetifolia,* respectively. The left part figure clearly shows which region the model is more focused on by using heatmap from 0 to 1. The closer the score is to 1, the darker the color and the more important the region considered by the model. The *p*-value was calculated using TOMTOM by comparing our iDNA-ABF learnt motifs with STREME motifs. The *p*-value in STREME was calculated by a one-sided binomial test. The motifs within the gray dashed anchor boxes were extracted for pair comparisons

*k*-mer integration (i.e., 3-mer + 6-mer) improves the model performance as compared to the single-scale *k*-mers (i.e., 3-mer, and 6-mer). To be specific, the model using the integration of 3-mer and 6-mer achieved the highest performance with the average ACC of 85.95% on all the datasets, which is 2.53 and 1.01% higher than that using 3-mer and

6-mer, respectively. This demonstrates that the information from different scales is complementary to each other for learning better feature representations.

Next, we further investigated why using multi-scale *k*-mer integration is more appropriate for discriminative information capturing. For this, we utilized attention mechanism to intuitively interpret the information our model learnt from two sequential scales—3-mer and 6-mer. We visualized the attention heatmap of the two scales in Fig. 3C and D, respectively. Note that the element in the heatmap represents the correlation degree of two positions along the sequences. Figure 3C shows the information our model learnt before and after training at 3-mer scale. As we can see, as compared to the initial model, the attention mechanism is more focused on the diagonal of the heatmap after training. This indicates that our model learns more local discriminative information as compared to that before training. Similarly, Fig. 3D illustrates the information our model learnt before and after training on the other sequential scale—6-mer. In contrast, this scale is more focused on global information after training. To this end, we can conclude that different scales of sequential patterns learn both local and global information, which might be complementary for the performance improvement.

In order to clearly demonstrate which sequential region is the most important for methylation prediction, we randomly selected three sequences from three species with different DNA methylation types, and applied the attention mechanism to identify key regions from these sequences. As can be seen in Fig. 3E–G (in left), for each sequence, our model identified different regions under different sequential scales. This further confirms that different scales capture different important information. For those identified regions, we further extracted and visualized the corresponding motifs using attention scores. Figure 3E–G (in right) shows the motifs learnt by our iDNA-ABF and that discovered by the conventional tool—STREME [28], respectively. As seen, our learnt motifs (highlighted with a gray-color window) almost match the STREME's motifs in each species. To quantitatively compare the motif similarity, we adopted TOMTOM [29] to calculate the similarity degree of two motifs, which is measured by *p*-value. The lower *p*-value indicates a higher degree of motif consistency. As can be seen in Fig. 3E–G, our motifs are highly similar to the STREME's motifs, suggesting that our model can learn conserved sequential characteristics.

### Our iDNA-ABF sufficiently explores genomic information in 5mC prediction across human cell lines

In this section, we analyzed how well our iDNA-ABF performs the methylation prediction across human cell lines. Since 5mC is one of the most well-studied methylation types in human genome, we selected the 5mC methylation to perform our method. We therefore constructed three new 5mC datasets corresponding to three human cell lines, including GM12878, K562, and HepG2, respectively. The details of the datasets can be seen in section "Datasets".

First, we discussed the impact of the length of methylated sequential regions for the 5mC methylation prediction. Therefore, for each cell line, we constructed four 5mC datasets, in each of which the 5mC sequences are 11, 41, 71, and 101 bp (base pairs) long, respectively. The details of the datasets are summarized in Additional file (Additional file 1: Table S2, Table S3, and Table S4). Figure 4A shows the model performance

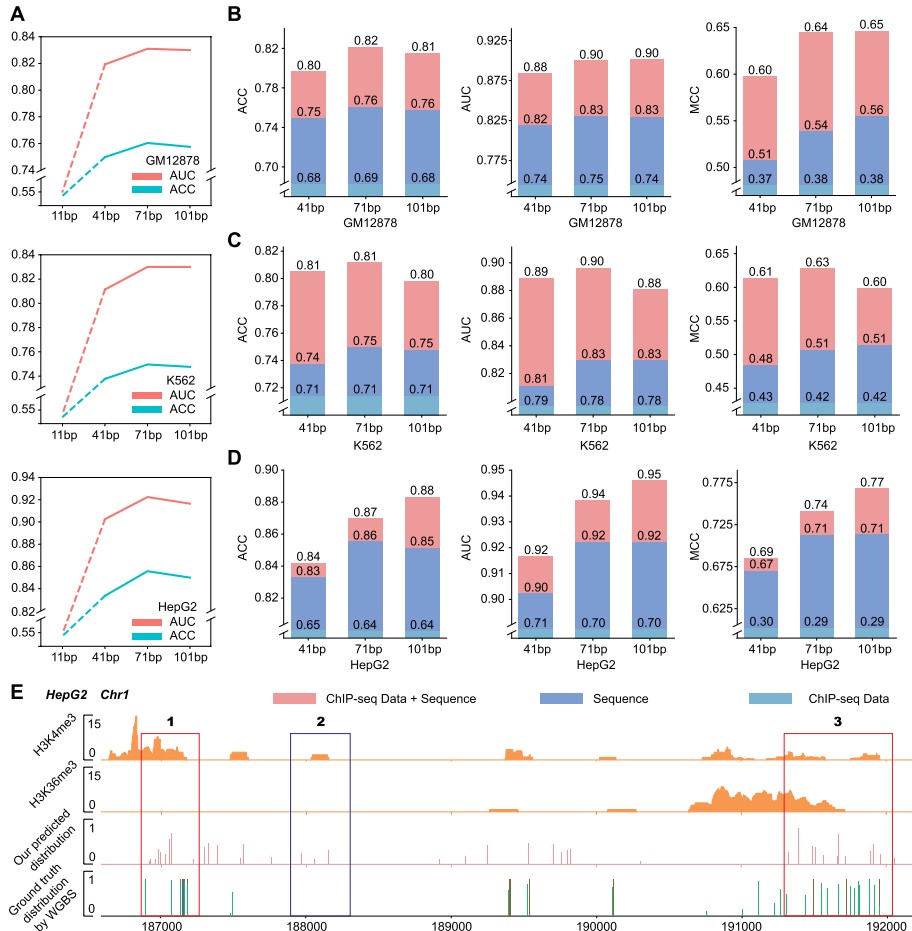

**Fig. 4** The 5mC prediction performance of our model on three human cell lines. **A** The ACC and AUC results of different human cell line datasets with different sequence lengths. **B–D** Performance of sequence data, ChIP-seq data, and integration of sequence and ChIP-seq data under different sequence lengths in three human cell lines, respectively. **E** The 5mC distributions predicted by our model and annotated by WGBS in a randomly selected genomic region (Chr1: 187000 - 192000, GRCh38). Note that the 5mC distribution is derived from HepG2 cell line

varied with different sequence lengths in the three cell lines. In the beginning, the model performance significantly improves as the sequence length increases, demonstrating that a longer sequence brings the model extra genomic contextual information. The peak is reached when the length is 71 bp. After that, the model performance gradually declines. Notably, the model trained with the sequences 11 bp long exhibits extremely poor performance, with the ACC of around 55%. The reason is that methylation-centered regions with the range of 11 bases are very similar between negative and positive samples. This further demonstrates that the methylations are strongly correlated with the upstream and downstream from the methylated regions.

As well known, 5mC methylation is one of the well-studied methylation types, backing supported by many NGS data, such as ChIP-seq data, and ATAC-seq data, etc. [30]. An interesting question is whether integrating the NGS data with sequence data can contribute to more accurate prediction. For this, we chose two histone modifications (HM) data, H3k4me3 and H3k36me3, which are reported to be closely associated with

5mC [31]. We trained and tested the models using (1) Sequence data only, (2) ChIP-seq data only, and (3) Sequence + ChIP-seq data on the three cell lines, respectively. The comparative results are shown in Fig. 4B–D. As we can see, the model trained with sequence data achieved remarkably better performance as compared to that trained with ChIP-seq data, leading by 10.4, 10.9, and 21.1% in the average ACC, AUC, and MCC in three cell lines under different sequence lengths. When combining ChIP-seq data with sequence data for model training, all the performance metrics are further improved, achieving the highest scores, with the improvement of 3.8, 5.2, and 8.1% on the average ACC, AUC, and MCC over the model trained with sequence data, demonstrating that the ChIP-seq data and sequence data are complementary to each other for the improved 5mC prediction.

### Application of iDNA-ABF for the 5mC methylation detection at genome scale

Considering real application scenario, it is important to measure the performance of our iDNA-ABF in detecting the 5mC distribution from the whole-genome scale. Thus, we predicted the methylation probability on a 5k-bp-long genomic region (Chr1: 187,000–192,000) from human genome (GRCh38) based on our iDNA-ABF model trained on HepG2. The prediction procedure is as follows. Firstly, we used a 71-bp-long window to screen the region. Secondly, the sequences that meet the following two requirements: (1) centered with base C and (2) centered with CPG patterns were picked out. Ultimately, the resulting sequences were submitted to our iDNA-ABF for prediction. Our model gives the predicted confidence of each site candidate.

Figure 4E illustrates two HM data distributions, the 5mC distribution predicted by our model, and the true 5mC distribution annotated by WGBS, respectively. As we can see from Fig. 4E, our predicted 5mC distribution is generally overlapped with the true 5mC site distribution. Moreover, the predicted 5mCs basically match with the two HM data, demonstrating that our predictions have the functional significance. Notably, we found that our model identified some regions (with blue frame, Fig. 4E) that are not identified by WGBS, but they matched well with the signal of H3K4me3 data. This implies that our model might discover potentially novel functional regions. Although our model also produces some false positives, from the perspective of sequential bins (here, we considered 100-bp region as a bin), the predicted 5mC region distribution is almost the same with the true 5mC region distribution. The results at least demonstrate that our model can perform well in locating 5mC regions. This could also be helpful for methylation research.

### Our iDNA-ABF has robust performance in 5mC prediction on unseen human cell lines

To analyze the predictive performance of iDNA-ABF in unseen cell lines, we conducted the cross-cell line validation. To be specific, we trained our model on one cell line and evaluated it on the other. Figure 5A shows the heatmap results in terms of four metrics, including ACC, MCC, SN, and SP, respectively. The vertical axis denotes training cell lines, while horizontal axis shows testing cell lines. As shown in Fig. 5A, our model achieved relatively stable ACC and MCC under the cross-cell line validation. Moreover, we can also see that when evaluated on the K562, our model trained on the GM12878 achieved the highest SN, yielding a relative improvement of 16% compared to the model

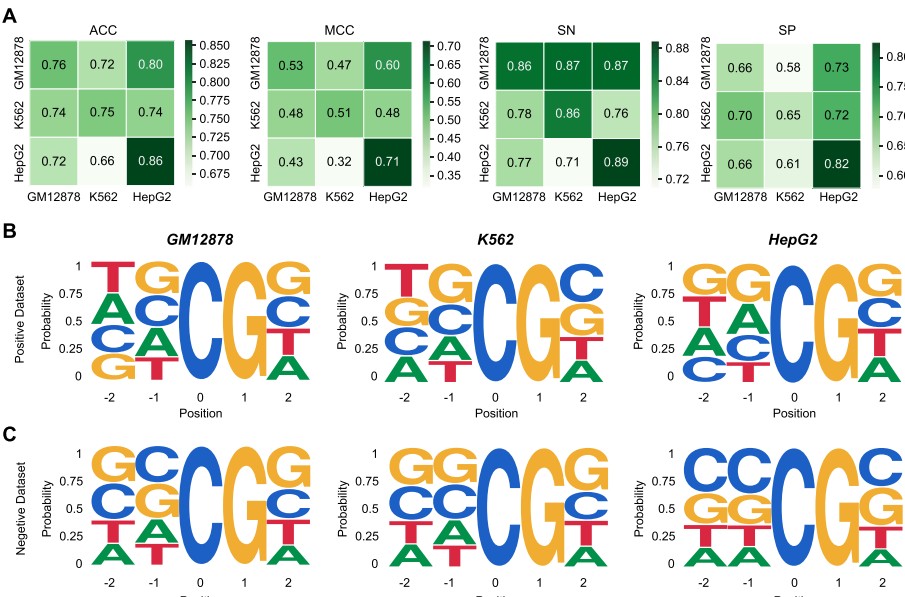

**Fig. 5** Performance in 5mC prediction on unseen cell lines. **A** The heatmap of cross-cell line validation in terms of different metrics, including ACC, MCC, SN, and SP, respectively. **B** Motif logos in central sequential regions of the positive datasets in three human cell lines, respectively. **C** Motif logos in central sequential regions of the negative datasets in three human cell lines, respectively

trained on the HepG2. For better explanation, we introduced the probability distribution analysis in methylated central regions of three human cell lines. Figure 5B and C show the probability distribution in the positive and negative samples in three cell lines, respectively. On the one hand, it can be seen from Fig. 5B that the positive motif logos of K562 are more similar to the GM12878 than HepG2 in position from −1 to 1. On the other hand, we observed from Fig. 5C that the negative motif logos of K562 are the same as the positive motif logos of GM12878 in position from −1 to 1, which can explain the lowest SP of our model trained on GM12878 while tested on K562. Furthermore, we found in Fig. 5A that our model in the heatmap of cross-cell line validation performs not that well in terms of SP. This might be that the negative motif logos among all three cell lines are quite different. To this end, via the cross-cell line validation results, we can conclude that our model has robust performance even for the unseen cell lines. This further explores the application value of our model.

### Our iDNA-ABF has good transfer learning ability to capture the specificity of methylated sequential patterns

The 5mC methylations mainly occur within the sequences with CpG patterns in human genome; actually in a few cases, the methylations are also detected within the CHH and CHG patterns (where H = A, C, or T). In order to find out whether different methylated sequential patterns are correlated with each other, we constructed extra CHG and CHH datasets for the three cell lines, respectively. It is worth noting that the number of sequences in CpG dataset is far more than that in CHG or CHH datasets. The details of the datasets are presented in Additional file (Additional file 1: Table S5).

To see whether our model has good transfer learning ability in detection of different methylation patterns, we firstly pretrained a model on the CpG dataset and fine-tuned it on the CHG or CHH datasets, yielding another model denoted as "transfer learning model". Moreover, we also trained a model directly with the CHG or CHH datasets for comparison, denoted as "baseline model". Both models were then evaluated with the same testing datasets of the CHG or CHH datasets. The performance on the two datasets is shown in Fig. 6A and B, respectively. As we can see, the performance of the "transfer learning model" is always superior to the baseline model, with the average AUC and AP increasing by 3.1 and 3.3% in three cell lines. The results demonstrate that our model has a good transfer learning ability; the pre-training mechanism can bring extra discriminative information from one specific pattern to benefit the prediction of the target patterns, thus improving the predictive performance.

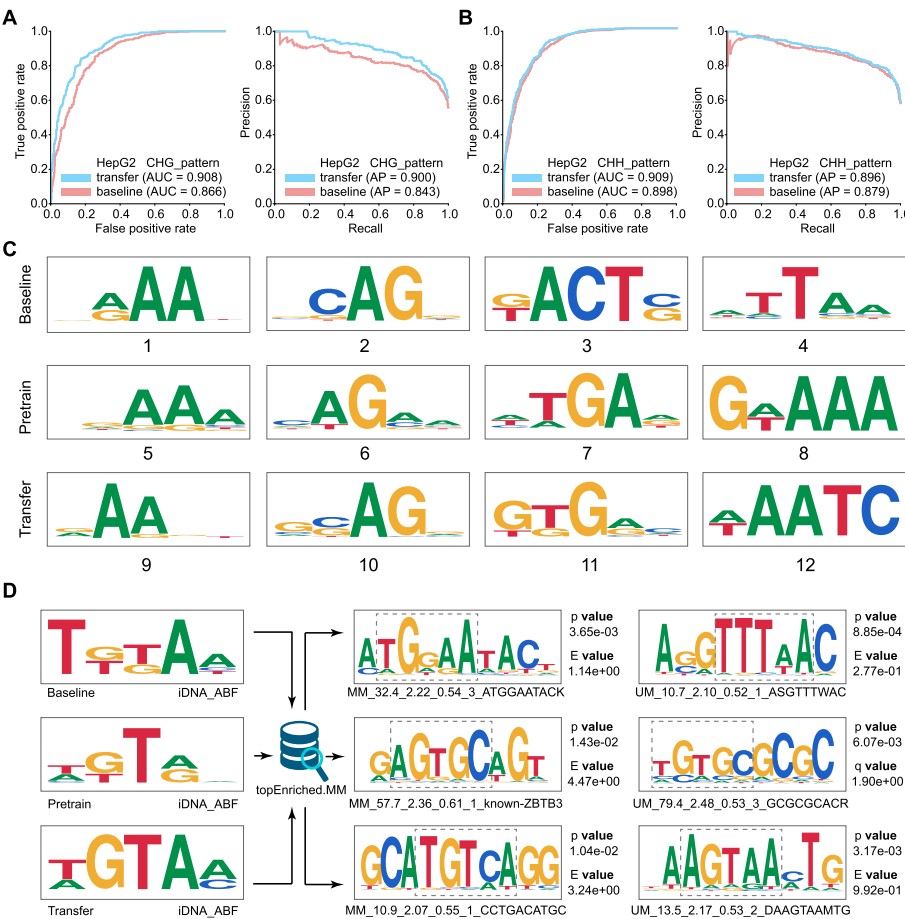

**Fig. 6** Transfer learning results and analysis of our model. **A** The ROC and PR curves of the baseline model and the transfer learning model on CHG dataset in HepG2 cell line. Note that the baseline model is trained with the CHG dataset while the transfer learning model is pretrained with CpG dataset and fine-tuned on the CHG dataset. **B** The ROC and PR curves of the baseline model and the transfer learning model in CHH dataset in HepG2 cell line. Note that the baseline model is trained with the CHH dataset while the transfer learning model is pretrained with the CpG dataset and fine-tuned on the CHH dataset. **C** The motifs learnt from three models, including baseline model, pretrained model, and transfer learning model, respectively. **D** The searching results using our learnt motifs against the topEnriched.MM database

To in-depth explain the possible reason regarding the performance improvement using transfer learning, we further analyzed and compared the motifs that were learned from the three models, including baseline model, transfer learning model, and pretrained model which is trained on CpG dataset only, respectively. The motif comparison results in HepG2 cell line are illustrated in Fig. 6C. Taking the 1st, 5th, and 9th motif figures as an example, we observed that the transfer learning model not only keeps some CpG patterns inherited from the pretrained model but also captures the specificity of the CHG patterns learned by baseline model. In addition, the transfer learning model can also discover some new patterns such as the 7th motif figure, which does not share similar patterns from the baseline model in the 3th motif figure.

### The motifs learnt from our models are biologically meaningful

Next, we further explored whether the motifs (or sequential patterns) learnt from our above three models (baseline model, pretrained model, and transfer learning model) are biologically meaningful. Accordingly, we searched the learnt motifs against topEnriched. MM, a public methylation database [32]. Interestingly, from Fig. 6D, we found that the motifs learnt by our models can significantly match with some functional motifs in the database, which were previously reported to be closely associated with methylation mechanisms. The results demonstrate that our model can accurately mine functional sequential characteristics; on the other hand, the newly discovered sequential motifs are also biologically meaningful, indicating the strong ability of our model in learning functional semantics between different sequential patterns.

### Discussions

We presented iDNA-ABF, a novel method for identifying DNA methylation by biological language learning solely based on genomic sequences. Our iDNA-ABF not only enables relatively accurate methylation prediction across species and across cell lines, but also builds the mapping from the sequential level to the functional level using explainable attention mechanism to study the in-depth DNA methylation mechanisms.

First, we investigated the predictive performance of our model to see how well and how stable it performs. Experimental results in 17 benchmark datasets covering three methylation types (4mC, 5hmC, and 6mA) in multiple species show that our model exhibits the consistently superior and robust performance as compared with the state-of-the-art sequence-based approaches. The ablation analyses reveal the importance of the adversarial training in the model performance. Particularly, the adversarial training in our training process alleviates the impact of large-scale parameters particularly on some small datasets and improves the generalization ability of our model across different species and methylation types. In addition, we also studied the impact of sequence length on the methylation prediction. The results show that the model performance generally improves with the increase of the input sequence length, demonstrating the upstream and downstream surrounding the methylated sequential regions are crucial to identifying DNA methylation sites. They might contain some degree of specificity information from the sequential perspective to help our model distinguish the methylation sites from non-methylation sites; on the other hand, we adopted DNABERT [33] for model construction, a powerful natural language learning model that was pretrained

with million-scale genomic sequence data. It enables our model to capture more sequential semantics from background genomes. The feature space visualization analysis results prove that our model learns more distinguishable feature representations as compared with existing predictors. Interestingly, by integrating ChIP-seq data such as histone modification data (e.g., H3K4me3 and H3K36me3) into our model, we observed that the performances are further improved, indicating that biological signals and sequence data are complementary for the improved prediction. The result could show the great potential of sequence data for DNA methylation prediction and other genomic functional analysis. We can imagine that, as for new cell line data, the NGS data is limited so that we cannot train an effective model. At least, the analyses provide a new way to build a more accurate and robust model by integrating the sequence data.

Secondly, the main feature of our iDNA-ABF is that we provide the interpretable analysis for DNA methylation prediction. The major problem of existing deep learning-based approaches is that they cannot well explain why their models are effective for the methylation prediction, since deep learning works as "black box". To address this problem, we did two major improvements for model construction. One is proposing the multi-scale sequence processing strategy for model training, and the other is introducing the attention mechanism for model analysis. Inspired by word segmentation in natural language learning, we utilized the multi-scale sequence processing strategy by segmenting DNA sequences with different scales (3mer and 6mer) of sequential patterns to represent "biological words". Furthermore, we adopted the attention mechanism to interpret what information our model learned from different scales of "biological words". Analyses demonstrate that the multi-scale strategy is capable of bringing more discriminative semantics information from both local and global levels, effectively overcoming the information lack at one single-scale and the information over-redundancy at all scales. Importantly, different sequential scales lead our model to learn different motifs. The results show that our learnt motifs from different scales are highly consistent with that by the conventional motif finding tool—STREME, demonstrating that our model is capable of discovering conserved sequential patterns. Next, the natural question is whether the sequential patterns learnt by our model are biologically meaningful or correlated with the methylations. To answer this question, we applied our model to the prediction of 5mC methylation across human cell lines. The reason to choose human 5mC methylation is that it has conserved methylated sequential patterns, such as CpG, CHH, and CHG; on the other hand, it is back supported by many NGS data, facilitating further functional validation analysis. We investigated the transfer learning ability of our model and the experimental results show that learning the knowledge from CpG methylation patterns can help the improved prediction of the other two methylation patterns (i.e., CHH, and CHG). This demonstrates that our model has a strong ability in learning the specificity of different methylation patterns. The results also imply the potential of our model in the discovery of other rarely occurred methylation patterns. Importantly, we found that by using transfer learning, our model can learn some new motifs and meanwhile keep the motifs in original methylation patterns. By searching our learnt motifs against a well-known methylation database—topEnriched.MM, we found that our motifs are significantly similar to some known methylation-related functional

motifs (see Fig. 6D). This also demonstrates that our model can learn different biological semantic information under different methylation patterns.

Ultimately, to verify the performance of our model in real application scenarios, we further applied our model to the detection of 5mC methylation within human genome. The experimental results in a randomly selected genomic region show that our model is capable of accurately detecting true DNA methylation regions (annotated by WGBS). Importantly, our model discovered some potential methylation regions, which are not detected by WGBS but are highly overlapped with the methylation-related histone modification data (e.g., H3K4me3). This demonstrates the strong ability of our model in the discovery of biologically meaningful sequential regions. It might be that the deep pretrained model helps us learn functional semantics from millions of background genome sequences.

## Conclusion

Altogether, our proposed deep biological language learning model achieves satisfactory performances in DNA methylation prediction. Importantly, we show the power of deep language learning in capturing both sequential and functional semantics information from background genomes. Moreover, by integrating the interpretable analysis mechanism, we have well explained what we learned, helping us build the mapping from the discovery of important sequential determinants to the in-depth analysis of their biological functions. However, there is still much room to improve. For example, in the construction of methylation prediction models, we only considered local sequential regions surrounding the methylation sites, in which the discriminative information might be limited to some extent. Studies [34] have demonstrated that there is a long-range interactive impact of gene regulations in genome, such as enhancer-promoter interaction. Therefore, exploring how the long-range sequence integrative information affects DNA methylation levels could be an important direction in future work.

## Methods

### Datasets

#### *Different species datasets*

A stringent dataset is fundamentally crucial for training effective and promising predictors. To further evaluate our proposed method with state-of-the-art methods, we choose the same benchmark datasets originally proposed by iDNA-MS [21]. The datasets consist of three main DNA methylation types, including seventeen datasets totally. Among seventeen datasets, *C. equisetifolia* (*4mC_C.equisetifolia*), *F. vesca* (*4mC_F.vesca*), *S. cerevisiae* (*4mC_S.cerevisiae*), and *Ts. SUP5-1* (*4mC_Ts.SUP5-1*) belong to 4mC. The 6mA contains *Arabidopsis thaliana* (*6mA_A.thaliana*), *Caenorhabditis elegans* (*6mA_C.elegans*), *Casuarina equisetifolia* (*6mA_C.equisetifolia*), *Drosophila melanogaster* (*6mA_D. melanogaster*), *Fragaria vesca* (*6mA_F.vesca*), *Homo sapiens* (*6mA_H.sapiens*), *Rosa chinensis* (*6mA_R.chinensis*), *Saccharomyces cerevisiae* (*6mA_S.cerevisiae*), *Tolypocladium sp SUP5-1* (*6mA_Tolypocladium*), *Tetrahymena thermophile* (*6mA_T.thermophile*), and *Xanthomonas oryzae PV. Oryzicola* (*Xoc*) *BLS256* (*6mA_Xoc.BLS256*). What is more, there are two 5hmC datasets from two species, including *H. sapiens* (*5hmC_H.*

*sapiens*) and *M. musculus* (*5hmC_M.musculus*). It should be noted that both positives and negatives are 41-base pair (bp) long and the sequence identity of the datasets is less than 80% using the CD-HIT [35] program, which is shown in Fig. 7A. The details of the training dataset and the validation dataset from seventeen species are given in Additional file (Additional file 1: Table S6).

### Human cell lines datasets

The 5mC methylation data of three human cell lines (K562, GM12878, hepG2) were collected from ENCODE portal (ENCSR765JPC, ENCSR890UQO, and ENCSR786DCL) [36], which provides the location information of three methylation patterns (CpG, CHG, and CHH) experimented by whole-genome bisulfite sequencing (WGBS). To construct a high-quality dataset, methylation sites with 100% methylated and 10−200× sequencing coverage were kept for positive samples, whereas methylation sites with 0% methylated and 0 sequencing coverage were selected as negative samples. The processed methylation sites located in promoter and gene body region were further mapped using annotation from GENECODE GRCh38. A promoter region is defined as the 1000-bp region upstream from the transcription start site

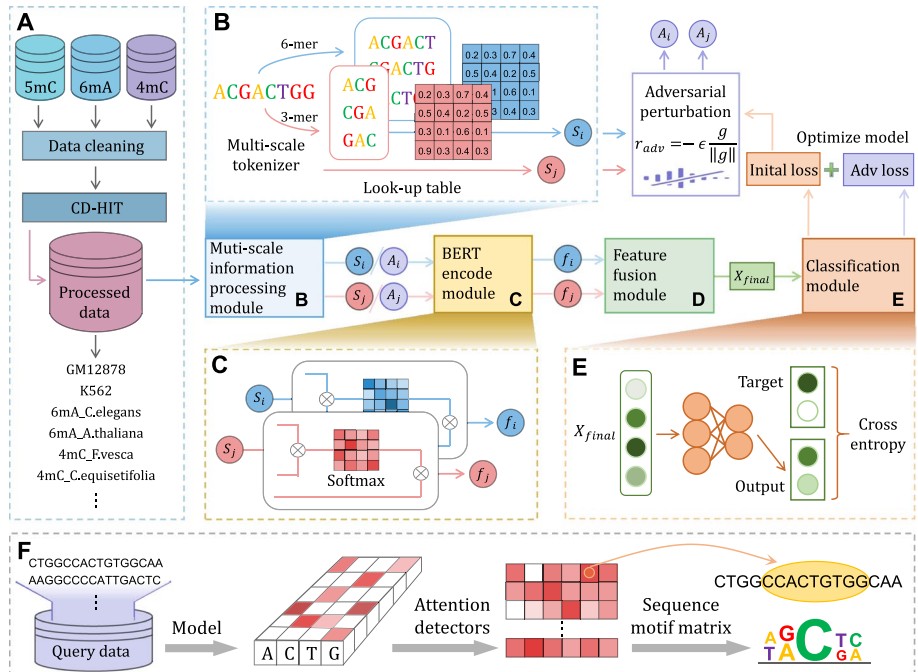

**Fig. 7** Overview of the proposed iDNA-ABF. **A** shows the DNA methylation dataset collection where different datasets belonging to three main DNA methylation types are reorganized into their training datasets and independent datasets. The overall architecture of our iDNA-ABF is presented in **B–E**. **B** Multi-scale information processing module, exploiting two scales (3-mer and 6-mer) of tokenizers separately to process the input sequence and adaptively obtain corresponding embeddings. **C** BERT encode module, using BERT encoders to extract high-latent feature representations. **D** Multi-scale extraction module, generating final output feature representations based on multi-scale embeddings. **E** Classification module, integrating binary classification probability values to make prediction. **F** The workflow of the interpretable analysis. In brief, our model uses attention mechanisms to extract and learn sequential motifs from query sequences

(TSS) of a gene. The number of each cell line processed dataset is shown in Additional file (Additional file 1: Table S7, Table S8, and Table S9). To evaluate the impact of sequence length on the prediction of methylations, the DNA sequences that 11, 41, 71, and 101-bp-long flanking the methylation sites were extracted from GRCh38, respectively. Similar to the different species datasets, the sequence identity of the human cell line datasets is also less than 80% with using the CD-HIT program.

### Description of the proposed iDNA-ABF

Figure 7 illustrates the overall architecture of our iDNA-ABF. Figure 7A shows the data set collection procedure, which is described in "Datasets" section. The workflow of iDNA-ABF is clearly seen in Fig. 7B–E, mainly consisting of four modules: (B) Multi-scale data processing module, (C) BERT encoder module, (D) Feature fusion module, and (E) Classification module. The prediction procedure is described as follows. In module B (see Fig. 7B), we exploit two scales of tokenizers (3-mer and 6-mer) separately to process the input sequence and adaptively learn corresponding embeddings. Due to the input sequences containing multifaceted features of various scales, we design a multi-scale architecture rather than using a single simple tokenizer, which may result in information loss. Afterwards, in module C (see Fig. 7C), the iDNA-ABF uses BERT encoders individually to extract different embeddings processed by tokenization. The iDNA-ABF then combines multi-scale embeddings based on BERT output in module D to generate the final evolutionary output feature. After that, in module E (see Fig. 7E), the model uses fully connected layers to predict whether the input sequence is methylated or not. Notably, we adopt adversarial training to enhance the robustness of the model and prevent early overfitting, which can be separated into two components: (1) adversarial perturbation, using cross-entropy loss from this propagation as the adversarial perturbation back to the network, and (2) adversarial optimization, obtaining the adversarial loss which is used to make backpropagation and optimize our model. Note that we describe the details of the four modules as follows.

### Multi-scale information processing module

In our model, we tokenize a DNA sequence with $k$-mer representations. In this way, each token is represented by $k$ bases, thus integrating richer contextual information for each nucleotide. For example, a given DNA sequence "ATGGCTG" can be tokenized to a sequence of two 6-mers: ATGGCT and TGGCTG. Different $k$ results in different token representations. In our work, we set $k$ as 3 or 6, and thus obtain two scales of token representations. The whole token table has $4^k+5$ tokens, consisting of all the permutations of $k$-mer as well as 5 special tokens: [CLS], [PAD], [UNK], [SEP], and [MASK], which stand for classification token, padding token, unknown token, separation token, and masked token, respectively.

### BERT-based encoder module

#### *Pre-training of the BERT model*

BERT is the first bidirectional language representation model based on the transformer proposed by [24]. Due to its powerful performance in language understanding against

many kinds of large corpus, BERT has been widely used in lots of NLP tasks. To better play the role of BERT, it generally will first be trained on a large background-related corpus with two pre-training tasks namely the masked language model and the next sentence prediction. Here we use a pretrained BERT model namely DNABERT [33], using the same architecture as the BERT base, which consists of 12 Transformer layers with 768 hidden units and 12 attention heads in each layer. Notably, since there is no direct semantic logic between DNA sequences, this domain pre-training adjusts the sequence length and enables the model to predict contiguous $k$ tokens adapting to a DNA sequence. Also, it uses the masked language model technique similar to the original BERT.

### Encoding process of the BERT

BERT is a transformer-based contextualized language representation model, which has been applied to many aspects of biology and has achieved many outstanding performances. The basic component of BERT consists of a multi-head attention mechanism, a feed-forward network, and the residual connection technique. To capture contextual information, BERT performs the multi-head attention mechanism based on the self-attention mechanism, which is described as follows:

$$\begin{cases} Q = XW^Q \\ K = XW^K \\ V = XW^V \end{cases} \tag{1}$$

$$Self - Attention(Q, K, V) = softmax\left(\frac{QK^T}{\sqrt{d_k}}\right)V \tag{2}$$

where $X \in R^{L \times d_m}$ is the output of the sequence embedding module. What is more, according to respectively linear layers $W^Q, W^K, W^V \in R^{d_m \times d_k}$, $X$ is transformed to the query matrix $Q \in R^{L \times d_k}$, key matrix $K \in R^{L \times d_k}$, and value matrix $V \in R^{L \times d_k}$, in which $L$ is the length of the input protein sequence, $d_m$ is the initial embedding dimension, and $d_k$ is the dimension of matrix $Q$, $K$, and $V$.

From the above base unit, the multi-head attention mechanism can be expressed as follows:

$$\begin{cases} Head_i = Self - Attention\left(XW_i^Q, XW_i^K, XW_i^V\right), i = 1, \dots, h \\ MultiHead - Attention(Q, K, V) = [head_1, head_2, \cdots, head_h]\, W^O \end{cases} \tag{3}$$

where $W_i^Q, W_i^K, W_i^V \in R^{d_m \times d_k}$ are the query, key, and value linear transformation layers of the $i^{th}$ head while $h$ is the number of heads. Then multi-head concatenates results of $h$ independent head with different sets of $\left\{W_i^Q, W_i^K, W_i^V\right\}$ and use a linear conversion layer $W^O$ to map the output dimension of the multi-head attention to the initial embedding dimension of the embedding module. The entire procedure is performed $L$ times, where $L$ represents the number of layers.

### Feature fusion module

In order to obtain the final output $h_M$ of two BERT parts, we combine the output $h_{kmer1}$ from the first scale input format layer and $h_{kmer2}$ from the second scale input format layer through a dimensional-wise fusion gate $F$. $F$ is accomplished by the sigmoid activation function to encode two parts of representation:

$$F = sigmoid(\ W_1 \cdot h_{kmer1} + W_2 \cdot h_{kmer2})\tag{4}$$

where $W_1$ and $W_2$ are trainable parameters of the fusion gate. Then the final vector representation output of a specific molecule $h_M$ is generated through F:

$$h_M = F \cdot h_{kmer1} + (1 - F) \cdot h_{kmer2}\tag{5}$$

### Classification module

Adversarial training [37] is a novel regularization method for classifiers to improve robustness to small, approximately worst-case perturbations. Here, because of the large parameters that BERT has, we use this strategy to prevent models from overfitting. Among lots of adversarial training methods, we use a variant of Fast Gradient Method (FGM) specific for text classification [38]. The cross-entropy loss function $L_{CE}$ is used to train the output module to improve the prediction performance as our base loss function. We define $p$ as the prediction probability, $y$ as the true label, $x$ as the input, $\theta$ as the parameters of the model, and $\varepsilon$ as one additional parameter. When applying this method, adversarial training adds the following term to the cost function:

$$\begin{cases} L_{CE}\big(p, y|x, \theta\big) = -y \log p - \big(1 - y\big) \log\big(1 - p\big) \\ L_{CE}\big(p, y|x + r_{adv}, \theta\big) \ where\ r_{adv} = \arg \min_{r, ||r|| \leqslant \varepsilon} L_{CE}\big(p, y|x, \hat{\theta}\big) \end{cases}\tag{6}$$

where $r$ is a perturbation on the input and $\hat{\theta}$ is a constant set to the current parameters of the model. Backpropagation algorithm should not be used to propagate gradients through the adversarial example construction process which means $\hat{\theta}$ is not consistent with $\theta$ in the Eq. (6). Then, in the training process, we minimize Eq. (6) for $\theta$ to obtain the worst-case perturbations $r_{adv}$ against the current model.

In the FGM method, we apply the adversarial perturbation to the extracted sequence embedding, rather than directly to the input. To define adversarial perturbation on the word embeddings, we denote relevant embedding of $k$-mer as $s$. Then we define the adversarial perturbation $r_{adv}$ on $s$ as

$$r_{adv} = -\varepsilon\ \frac{g}{||g||_2} where\ g = \nabla_s\ L_{CE}\big(p, y|s, \theta\big)\tag{7}$$

To train a robust model, we define a new adversarial loss based on the adversarial perturbation defined in Eq. (6), which is formulated as follows:

$$L_{adv}(\theta) = -\frac{1}{N} \sum_{n=1}^{N} L_{CE}\big(p_n, y|s_n + r_{adv,n}, \theta\big)\tag{8}$$

where $N$ is the number of batch size. In our work, adversarial training is to minimize the $L_{adv}$ based on cross-entropy loss with stochastic gradient descent.

### Performance metrics

In this study, we evaluate the performance of our iDNA-ABF and other existing methods with the following four commonly used metrics: Accuracy (ACC), Matthews' correlation coefficient (MCC), Sensitivity (SN), and Specificity (SP). The formulas of these metrics are described as follows:

$$\begin{cases} ACC = \frac{TP+TN}{TP+FN+TN+FP} \\ MCC = \frac{TP\times TN - FP\times FN}{\sqrt{(TP+FP)(TP+FN)(TN+FP)(TN+FN)}} \\ Sensitivity(SN) = \frac{TP}{TP+FN} \\ Specificity(SP) = \frac{TN}{TN+FP} \end{cases} \tag{9}$$

where TP, FN, TN, and FP represent the number of true positive, false negative, true negative, and false positive samples, respectively. ACC and MCC are both used to measure the overall performance of the model. SN refers to the proportion of true methylated samples correctly predicted by a predictive model, and SP measures the proportion of non-methylated samples correctly predicted by the model. Moreover, the ROC (receiver operating characteristic) curve and PR (precision-recall) curve [39] are used to intuitively evaluate the overall predictive performance of the model. AUC and AP denote the area under ROC curve and that under the PR curve, respectively [39]. They are further used to quantitatively measure the overall performance of the model. Altogether, the higher these metrics are, the better the model is.

### Supplementary Information

---

Additional file 1: Table S1. Performances of iDNA-ABF and the state-of-the-art methods on 17 benchmark datasets across species and methylation types. Table S2. Performance of our iDNA-ABF under various methylation patterns varied with different sequence lengths in three human cell lines. Table S3. Performance of our iDNA-ABF using ChIP-seq data varied with different sequence lengths in three human cell lines. Table S4. Performance of iDNA-ABF using ChIP-seq data + sequence data varied with different sequence lengths in three human cell lines. Table S5. The transfer learning performance of our model varied with different sequence lengths in three human cell lines. Table S6. The statistics of 17 benchmark datasets with three methylation types in various species. Table S7. The statistics of raw data with different methylation patterns in three human cell lines. Table S8. The statistics of the data in three human cell lines after sequence similarity reduction using CD-HIT. Table S9. The statistics of training and testing data under different methylation patterns in three human cell lines. Table S10. Performance comparison of different scales as the input to train the model in various species. Table S11. Performance comparison with the 5mC methods on cancer cell line Encyclopedia (CCLE). Table S12. Performance of iPromoter-5mC in three human cell lines. Table S13. Training parameters of our model on 17 benchmark datasets. Table S14. Performance of our iDNA-ABF for the SNP classification. Table S15. Performance of the multi-task model. Table S16. Performance of the regression model built on ChIP-seq data. Figure S1. The ROC curves on benchmark datasets. Figure S2. The PR curves on benchmark datasets. Figure S3. The UMAP visualization results on benchmark datasets. Figure S4. The SN, SP, and AUC of the models with and without adversarial training on 17 benchmark datasets with the independent test. Figure S5. Taxonomy tree and accuracy for eleven species in 6mA dataset. Figure S6. The ROC and PR curves of our baseline model and our model transferring from CpG pattern on the other two pattern datasets (i.e., CHG, CHH) in the cell line GM12878 and K562. Figure S7. The regression result of signal prediction in the regression model built on ChIP-seq data. Supplementary methods.

Additional file 2. Review history.

---

### Review history
The review history is available as Additional file 2.

**Peer review information**

**Authors' contributions**

J.J. conceived iDNA-ABF. J.J., Y.Y., and Z.L. performed the experiments and data analysis. X.Z. and Y.D. constructed the cell line datasets. Y.J. constructed the web server. J.J., R.W., Y.Y., C.P., R.S., Q.Z., K.N., and L.W. wrote, revised, and contributed to the final manuscript. L.W. and K.N. designed the study and supervised the project. All authors read and approved the final manuscript.

**Funding**

The work was supported by the Natural Science Foundation of China (Nos. 62071278, and 62072329). K.N. is supported by grants-in-aid for scientific research (22K06189), JSPS.

**Availability of data and materials**

The 5hmC site containing sequences for *H. sapiens* and *M. musculus* were collected from NCBI Gene Expression Omnibus (GEO) database under accession number GSE127906 [40].

The 6mA site data for 11 species (*Arabidopsis thaliana* (*A. thaliana*), *Caenorhabditis elegans* (*C. elegans*), *Casuarina equisetifolia* (*C. equisetifolia*), *Drosophila melanogaster* (*D. melanogaster*), *Fragaria vesca* (*F. vesca*), *H. sapiens*, *Rosa chinensis* (*R. chinensis*), *Saccharomyces cerevisiae* (*S. cerevisiae*), *Tolypocladium sp SUP5-1* (*Ts. SUP5-1*), *Tetrahymena thermophile* (*T. thermophile*), and *Xanthomonas oryzae pv. Oryzicola* (*Xoc*) *BLS256* (*Xoc. BLS256*)) were obtained from the MethSMRT database (http://sysbio.gzzoc.com/methsmrt/) with accession numbers SRP145409 [41] under BioProject PRJNA450482, MDR database (http://mdr.xieslab.org), GEO database under the accession number GSE104475 [42] and NCBI Genome database SRA: SRX1424851 and SRX1423750 in NCBI project SRA: PRJNA301527 [43], respectively.

The 4mC site data for 4 species (*C. equisetifolia*, *F. vesca*, *S. cerevisiae*, and *Ts. SUP5-1*) were obtained from the MDR database (http://mdr.xieslab.org) and MethSMRT database (http://sysbio.gzzoc.com/methsmrt/).

The human cell line dataset GM12878, K562, and HepG2 were obtained from ENCODE portal (ENCSR765JPC, ENCSR890UQO, and ENCSR786DCL) [36, 44]. The ChIP-seq data H3K4me3 and H3K36me3 were also obtained from ENCODE portal (ENCSR668LDD, ENCSR000DWB, ENCSR057BWO, ENCSR000DRW, ENCSR575RRX, ENCSR000DUD) [36].

To facilitate the use of our method, we established a code-free, interactive, and non-programmatic web interface of iDNA-ABF at https://server.wei-group.net/idnaabf, which can lessen the programming burden biological and biomedical researchers. Besides, the benchmarking datasets and our source code were also available at this server. In addition, our source code is also available at https://github.com/FakeEnd/iDNA_ABF under MIT license and at Zenodo [45, 46].

## Declarations

**Ethics approval and consent to participate**

Not applicable.

**Consent for publication**

Not applicable.

**Competing interests**

The authors declare that they have no competing interests.

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

## 