## [Additional file 2. Review history. · Genome Biology]

Review History

First round of review

Reviewer 1

Were you able to assess all statistics in the manuscript, including the appropriateness of statistical tests used? No

Were you able to directly test the methods? Yes

Comments to author:

In this study, the author proposed iDNA-ABF, an interpretable end-to-end deep learning model that uses DNA sequences only to predict the DNA methylation. Interestingly, they use the multi-scale strategy to capture key sequential key information that are associated with the functional motifs. Benchmark results show that the proposed method achieves the state-of-the-art performance. Moreover, they also adopt novel adversarial training which improves the accuracy of the model. The interpretable part in the study is fascinating and detailed. Overall, the work in this paper is novel and the manuscript is well-written as well. However, I have the following concerns that need to be addressed.

Major issues:

1. A server may need to be provided to users, which can be convenient for prediction and comparison in the following work, for example the iDNA-MS (<http://lin-group.cn/server/iDNA-MS/>).
2. The proposed iDNA-ABF shows good performance on 4mC, 5hmC, and 6mA predictions. However, 5mC is another important DNA methylation in common human diseases. So, to make the method more general to use, I wonder if the proposed method works well on other 5mC relevant datasets.
3. The authors only randomly select three sequences from three different types of DNA methylation datasets in section visualization of attention scores to mine important regions. I wonder if making the overall statics on whole datasets rather than single sequence may be more convinced. They should sum the scores and regularize according to the position.
4. Fig 7 shows the region which the model is more focused on. So, since the DNA sequence long sequence information and structural variants, I'm curious about whether distantly-located sequence features can be detected by short k-mers and whether the length of the input sequence can affect the final results.
5. In terms of seventeen datasets, the authors should consider to train a general pretrain model and then finetune on specific dataset or species? Actually, some recent studies have adopted this pipeline when using more than one dataset, such as BERT-6mA.
6. In discussions and conclusions, more insightful discussions should be given. For example, why does adversarial training enhance the accuracy of the model? What's more, authors should further discuss the possibility of using more handcrafted features as multi-information.

Minor issues:

The manuscript is overall well written, but there are some grammar errors in this manuscript. Below is a non-exhaustive list.

- p.4 l.39: "but also it plays key roles" -> "but also plays a key role"
- p.6 l.11: "sufficiently capture the different scale" -> "sufficiently captures the different scale"
- p.6 l.43: "Due to the input sequences contain multifaceted features of various scales" -> "Due to the input sequences containing multifaceted features of various scales"
- p.7 l.41: "mechanism and interpretation result" -> "mechanism and interpretation results"
- p.14 l.3: "focused on with using heatmap from 0-1" -> "focused on using heatmap from 0 to 1"

Reviewer 2

Were you able to assess all statistics in the manuscript, including the appropriateness of statistical tests used? No

Were you able to directly test the methods? No

Comments to author:

In the work, Junru et al., develop a multi-scale deep biological language learning model using pre-trained BERT for DNA methylation prediction. In general, this is a solid work. In particular, the concept of biological language learning is quite interesting. They also present some interesting results that the sequential patterns learnt from their model match up with some functional motifs. Moreover, their model shows promising performance via across-species validation. It would be more interesting and valuable to researchers after addressing the following problems:

Major comments:

(1) For data construction, they used several benchmark datasets, up to 17, covering three methylation types, including 5hMC, 4mC, and 6mA, across different species. The datasets are used as golden datasets for performance evaluation in few studies. Firstly, why to choose these specific species? Moreover, I notice, as for 5hmC and 4mC methylation types, most of the species datasets are not available. It would definitely influence the results to see how their model perform across different methylation types in the same species. Additionally, the dataset size in some species (like H.sapiens and M.musculus in 5hMC) are too small to train a robust deep learning model. Thus, to this end, it is highly recommended to reconstruct the datasets. Perhaps focusing on human genome with different cell lines would be a better option.

(2) Case study that performs their proposed method in unseen genome or species to detect DNA methylations is highly recommended to add in results section. It would be interesting to see how their predictions match with true methylation distribution generated from NGS.

(3) For feature representation visualization results, it would be more promising if they present the comparative results between their method and iDNA-ABT on all the seventeen datasets. At least, for fair comparison, they should perform the feature visualization comparison on different methylation types in the same species.

(4) In this study, they highlight one of their major contributions of their proposed model is that they can use sequence as input to train a good model. I agree that sequence information is quite important. Using sequences only can accelerate the application of the model. However, it would be also interesting how the performance change if they can integrate NGS data, such as Chip-seq data, etc.

(5) I wonder if a generic model can be trained to accurately predict all the methylation types of interest, not just the three major methylation types. If so, it would greatly increase the value of the work from reader side.

(6) They claim that their model can learn the biological language and performs well on methylation data. It would be interesting to see if the proposed method performs as well on other genetic variant detection.

Minor comments:

(1) The description of the proposed method iDNA-ABF should be placed in "Methods" section, rather than "Results" section.

(2) The figures in the main text should be further improved. The text in most of figures are not clear to see, such as Figures, 2,3,4,and 5.

(3) For the description of the iDNA-ABF, they used the uppercase ABCD to order each module in main text, but in figure 1 and the legend, the lowercase abcd... is used. They should keep them consistent.

(4) The English of this manuscript should be further polished.

(5) For result reproductivity, the hyper parameters should be given.

(6) In supplementary materials, I didn't find the results they mentioned in main text.

Reviewer 3

Were you able to assess all statistics in the manuscript, including the appropriateness of statistical tests used? No

Were you able to directly test the methods? No

Comments to author:

The authors proposed iDNA-ABF, a deep biological language learning model for the accurate prediction of multiple DNA methylations based on DNA sequence information only. They said that their model outperforms state-of-the-art methods on seventeen benchmark datasets across different species by a large margin, demonstrating the applicability, scalability, and reliability of our promising

approach to identifying DNA methylation, but I am seriously concerned with novelty of this paper and with the superiority of their proposed mode.

(1) The authors focused on 4-methylcytosine (4mC), 5-hydroxymethylcytosine (5hmC), and N6-methyldeoxyadenosine (6mA) are three DNA methylation types. They did not make clear whether

a) they constructed three differently trained models to prediction of each type or they three times repeated a binary classification method for each methylation type.

b) they constructed 3-class classification method.

I guess they adopt (a), but (a) has little novelty because many scientists published the binary classification methods of DNA methylations.

(2)

The authors need to investigate latest state-of-the-art predictors and are required to show some advantages over the latest predictors. It is important to compare their method with some latest predictors including

Li Z, Jiang H, Kong L et al. Deep6mA: A deep learning framework for exploring similar patterns in DNA N6-methyladenine sites across different species, PLOS Computational Biology 2021;17:e1008767.

Pian C, Zhang G, Li F et al. MM-6mAPred: identifying DNA N6-methyladenine sites based on Markov model, Bioinformatics 2019;36:388-392.

Sho Tsukiyama, et al. BERT6mA: Prediction of DNA N6-Methyladenine site using deep learning-based approaches. Briefings in Bioinformatics 23:1-15, 2022.

(3)

In Fig. 7, the authors visualized the attention scores of 4mC_S.cerevisiae, 5hmC_M.musculus, and 6mA_A.thaliana. In Fig. 8 they performed motif matching of 4mC Tolypocladium, 5hmC H.sapiens and 6mA C. equisetifolia/F.vesca. We like to see the consistency between the attention analysis and STREAME, but they used different species, which makes it difficult to see their consistency. They should conduct the sequence pattern matching by using the same species between Fig 7 and Fig 8.

(4)

Since the attention analysis of DNA methylation has already been carried, they need to briefly discuss their attention analysis based on other works.

For example, Sho Tsukiyama, et al. BERT6mA: Prediction of DNA N6-Methyladenine site using deep learning-based approaches. Briefings in Bioinformatics 23:1-15, 2022.

(5)

In Fig. 4, to demonstrate the effectiveness of adversarial training, the authors need to compute the metrics (SN, SP, ACC, AUC, MCC,...) scores on the training datasets. An independent test is necessary to demonstrate the superiority of the adversarial training.

Responses to Reviewer #1

Comments:

A server may need to be provided to users, which can be convenient for prediction and comparison in the following work, for example the iDNA-MS (<http://lin-group.cn/server/iDNA-MS/>).

Author's response:

We appreciate the Reviewer's comment. We have accordingly set up a user-friendly, interactive, code-free web interface as the implementation of iDNA-ABF, which is now freely accessible at <https://server.wei-group.net/idnaabf/>. Users can simply input their DNA sequence data in standard format (i.e., FASTA) to get prediction results. In addition, all the datasets used in this work can be also downloaded from our server. The source code of our method can be accessed via our GitHub link: https://github.com/FakeEnd/iDNA_ABF.

Comments:

The proposed iDNA-ABF shows good performance on 4mC, 5hmC, and 6mA predictions. However, 5mC is another important DNA methylation in common human diseases. So, to make the method more general to use, I wonder if the proposed method works well on other 5mC relevant datasets.

Author's response:

We appreciate this comment. We accordingly constructed the 5mC methylation datasets in different human cell lines, and conducted a series of comparative experiments and analysis. As for the prediction of 5mC methylation, we achieved satisfactory performance. As compared to iPromoter-5mC, which is the state-of-the-art 5mC predictor, our model leads by 2-3% in all metrics, such as ACC and MCC, etc. More details of the performances can be seen in the Additional file (Table S2 and Table S12). Important, we found that our model performs robust across different cell lines, which demonstrates the adaptability of our deep learning model in learning the specificity of cell lines. To further validate the superiority of our model, we also compared it with the existing predictors on other benchmark datasets collected from cancer cell line Encyclopedia (CCLE). The comparative performance is presented in Table S11, in which our model also shows better performance.

Comments:

The authors only randomly select three sequences from three different types of DNA methylation datasets in section visualization of attention scores to mine important regions. I wonder if making the overall statics on whole datasets rather than single sequence may be more convinced. They should sum the scores and regularize according to the position.

Author's response:

We appreciate this suggestion. In our revision, to interpret what our model learns, we visualized the motifs that our model learns based on the whole datasets using attention scores. Via the motif logo analysis, we can provide more insights to explain what information and what patterns our model emphatically captures from the datasets. In particular, we found that our model has a good transfer learning ability to learn and capture the specificity of different methylation patterns and generate some new motifs across different pattern datasets, which are then proven to be closely associated with known functional motifs. The other motif and interpretable analysis results can be found in our "Results" section and more insights discussion regarding this aspect can be found in "Discussions" section in the main text of our revision.

Comments:

Fig 7 shows the region which the model is more focused on. So, since the DNA sequence long sequence information and structural variants, I'm curious about whether distantly-located sequence features can be detected by short k-mers and whether the length of the input sequence can affect the final results.

Author's response:

We appreciate this comment. Since the sequences are fixed in our original species datasets, we instead constructed 5mC methylation datasets in different human cell lines. In particular, we curated the 5mC sequences in various length settings (i.e., 11bp, 41bp, 71bp, 101bp) and further carried out experiments to explore how the sequence length affects the predictive performance. We presented some important analysis result in the main text of our revision as follows

“Figure 4A shows the model performance varied with different sequence lengths in the three cell lines. At beginning, the model performance significantly improves as the sequence length increases, demonstrating that longer sequence brings the model extra genomic contextual information. The peak is reached when the length is 71 bp. After that, the model performance gradually declines. Notably, the model trained with the sequences with 11 bp long exhibits the extremely poor performance, with the ACC of around 55%. The reason is that methylation-

centered regions with the range of 11 bases are very similar between negative and positive samples. This further demonstrates that the methylations are strongly correlated with the upstream and downstream from the methylated regions.”

Comments:

In terms of seventeen datasets, the authors should consider to train a general pretrain model and then finetune on specific dataset or species? Actually, some recent studies have adopted this pipeline when using more than one dataset, such as BERT-6mA.

Author’s response:

We appreciate the Reviewer’s comment. Due to the diversity of methylation types and species in the benchmarking datasets and the limited amount of data, we did not consider the pretraining technique on the species datasets in our previous version. Of course, the pretraining technique is a good idea to facilitate the generalization of the model, so we pretrained and finetuned our model among different DNA methylation patterns in human cell lines. In particular, in the section of "Our iDNA-ABF has a good transfer learning ability to capture the specificity of methylated sequential patterns", we pretrained our model on the large dataset of CpG patterns, and then finetuned on other small datasets of CHG and CHH patterns. This is some kind of “Transfer Learning” actually. The results show that our performance is significantly improved by transfer learning, demonstrating that the pretrained model contain more discriminative information. More importantly, we demonstrate that transfer learning enables us to learn and capture the specificity of different methylation patterns and generate some new motifs across different pattern datasets, which are then proven to be closely associated with known functional motifs.

Comments:

In discussions and conclusions, more insightful discussions should be given. For example, why does adversarial training enhance the accuracy of the model? What's more, authors should further discuss the possibility of using more handcrafted features as multi-information.

Author’s response:

Thank the reviewer for such a good suggestion. We totally agree with the importance to provide the insightful discussion. We have rewritten the “Discussion” section in our revision. We have accordingly discussed both adversarial learning and multiple information in depth as follows, "*In this work, to train a robust model for DNA methylation prediction, we use adversarial training strategy in our training process to alleviate the impact of large-scale parameters particularly on some small datasets, and improve the generalization ability of our model across different*

datasets." and "Moreover, after introducing Chip-seq data, the performances of our model are further improved, indicating that the biological data and sequence data are complementary in DNA methylation prediction". Also, more other valuable discussion can be seen in this section as well.

Comments:

The manuscript is overall well written, but there are some grammar errors in this manuscript. Below is a non-exhaustive list.

Author's response:

Thank the reviewer to point out the question. We have fixed all the language problems as pointed out by the reviewer. Moreover, we have checked the whole article carefully to ensure no other similar issues.

Responses to Reviewer #2

Comments:

For data construction, they used several benchmark datasets, up to 17, covering three methylation types, including 5hMC, 4mC, and 6mA, across different species. The datasets are used as golden datasets for performance evaluation in few studies. Firstly, why to choose these specific species? Moreover, I notice, as for 5hmC and 4mC methylation types, most of the species datasets are not available. It would definitely influence the results to see how their model perform across different methylation types in the same species. Additionally, the dataset size in some species (like H.sapiens and M.musculus in 5hMC) are too small to train a robust deep learning model. Thus, to this end, it is highly recommended to reconstruct the datasets. Perhaps focusing on human genome with different cell lines would be a better option.

Author's response:

We appreciate the reviewer's suggestion. The reason we chose the datasets for performance comparison is that they are well applied to several studies, including "Lin et al., 2020, iScience",

and “Yu *et al.*, 2021, *Bioinformatics*”, etc. The limitation of the datasets is also obvious; as the reviewer mentioned, the number of species is not uniform for different methylation types, and thus little information can be extracted across species. In addition, the size of data sets across species is not consistent. Most of them are small data sets.

In our revision, we instead constructed 5mC methylation datasets in different human cell lines, since 5mC is one of the most well-studied methylation types in human genome. Specifically, we constructed three new 5mC datasets corresponding to three human cell lines, including GM12878, K562 and HepG2, respectively. Moreover, for each cell line, we curated the 5mC sequences under various length settings (i.e., 11bp, 41bp, 71bp, and 101bp) as well, and further explored how the sequence length affects the predictive performance. We list some important conclusions derived from the main text as follows:

*“...we applied our model to the prediction of 5mC methylation across human cell lines. The reason to choose human 5mC methylation is that it has conserved methylated sequential patterns, such as CpG, CHH, and CHG, etc.; on the other hand, it is back supported by many NGS data, facilitating further functional validation analysis. We investigated the transfer learning ability of our model and the experimental results show that learning the knowledge from CpG methylation patterns can help the improved prediction of the other two methylation patterns (i.e. CHH, and CHG). This demonstrates that our model has strong ability in learning the specificity of different methylation patterns. The results also imply the potential of our model in the discovery of other rarely occurred methylation patterns. Importantly, we found that using transfer learning, our model can learn some new motifs and meanwhile keep the motifs in original methylation patterns. By searching our learnt motifs against a well-known methylation database – topEnriched.MM, we found that our motifs are significantly similar with some known methylation related functional motifs (see **Figure 6D**). This also demonstrates that our model can learn different biological semantic information under different methylation patterns.”*

Comments:

Case study that performs their proposed method in unseen genome or species to detect DNA methylations is highly recommended to add in results section. It would be interesting to see how their predictions match with true methylation distribution generated from NGS.

Author’s response:

We appreciate the reviewer’s comments. In our revision, we have accordingly added some analysis to study the DNA methylation prediction performance of our method in unseen genome

and under cross-species, including the section "Our iDNA-ABF explores the methylation conservation cross species at sequential level" and section "Our iDNA-ABF has robust performance in 5mC prediction on unseen cell lines". The results of the sections demonstrated the robustness and generalization of our model. Moreover, in the section "Application of iDNA-ABF for the whole-genome identification of 5mC ", we performed our iDNA-ABF on a 5k bp long genomic region randomly derived from Chromosome 1 of human genome, ranging from 187,000 to 192,000 and the results (see Figure R1) show that our model can perform well in locating 5mC regions. Importantly, our model discovered some potential methylation regions, which are not detected by WGBS but are highly overlapped with the methylation related histone modification data (e.g., H3K4me3).

Figure R1. The 5mC distributions predicted by our model and annotated by WGBS in a randomly selected genomic region (Chr1: 187000 - 192000, GRCh38). Note that the 5mC distribution is derived from HepG2 cell line.

Comments:

For feature representation visualization results, it would be more promising if they present the comparative results between their method and iDNA-ABT on all the seventeen datasets. At least, for fair comparison, they should perform the feature visualization comparison on different methylation types in the same species.

Author's response:

Thank the reviewer for pointing out this question. Because the result text is limited, we ignored this problem at the beginning. We performed the feature visualization comparison between iDNA-ABT and our model as shown in Figure R2. The other results of the seventeen datasets can be seen in the Additional file (Figure S3). The results demonstrate that our model learns better feature representations from different classes samples in different species, possibly due to the well pre-trained model in our model construction, helping us capture more high-latent contextual semantics information from millions of background genomic sequences.

Figure R2. The feature space distribution (with UMAP visualization) of iDNA-ABF and iDNA-ABT in *5hmC_M.musculus*, *4mC_C.equisetifolia*, *6mA_C.equisetifolia*, and *6mA_F.vesca*, respectively.

Comments:

In this study, they highlight one of their major contributions of their proposed model is that they can use sequence as input to train a good model. I agree that sequence information is quite important. Using sequences only can accelerate the application of the model. However, it would be also interesting how the performance change if they can integrate NGS data, such as Chip-seq data, etc.

Author's response:

We appreciate the reviewer's advice Integrating ChIP-seq data (such as histone modifications, etc.) with the DNA sequence data is a great way to explore the relationship between histone modifications and DNA sequences. By incorporating ChIP-seq data, we can study DNA methylation from multiple perspectives. Therefore, we explored different ways (with different input sequence length setting, such as 41bp, 71bp, and 101bp) to incorporate ChIP-seq data, and found that integration of Chip-seq data with DNA sequence data greatly improved the predictive performance on all three cell line datasets. We demonstrate that biological signals and sequence data are complementary for the improved prediction. The results show the great potential of sequence data for DNA methylation prediction and other genomic functional analysis. The other details of result analysis can be found in the section "Our iDNA-ABF sufficiently explores genomic information in 5mC prediction cross human cell lines".

Comments:

I wonder if a generic model can be trained to accurately predict all the methylation types of interest, not just the three major methylation types. If so, it would greatly increase the value of the work from reader side.

Author's response:

We appreciate the reviewer's comment. We think this is a good idea, and accordingly constructed a multi-task learning model for the generic prediction of different DNA methylations. The other one is to distinguish which methylation type it is. The details of the proposed multi-task learning can be found in Additional file ("Supplementary Methods"). Specifically, we choose three methylation types in human genome, including 5hmC, 6mA, and 5mC. However, the performances of the generic model on the three methylation types are all lower than the model only trained on their own methylation type dataset. And the detailed experimental results can be seen from the Additional file (Table S15). We think the reason may be that different methylations have their own specific patterns and motifs which make the model not easy to make accurate predictions.

Comments:

They claim that their model can learn the biological language and performs well on methylation data. It would be interesting to see if the proposed method performs as well on other genetic variant detection.

Author's response:

We appreciate the reviewer's comment. In the beginning, we just make a binary classification between mutated and unmutated sequences using the training dataset and testing dataset in this website (<https://krishna.gs.washington.edu/download/CADD-development/v1.6/validation/gnomad/>). However, the result is not well as shown in the Additional file (Table S14). To the best of our knowledge, the methods to detect SNP such as DeepVariant use the sequence alignment information, instead of using sequence information. Furthermore, in order to perform more valuable and exploratory SNP variation experiments, we used the model trained based on DNA methylation dataset to predict the methylation level of the sequence before and after the mutation to see whether methylation would have an impact on the mutation. The result is that there is no significant drop in accuracy where the reduction value is just only 0.32%. Thus, even if the sequence has a SNP mutation, our model can be robust for the DNA methylation prediction.

Comments:

The description of the proposed method iDNA-ABF should be placed in "Methods" section, rather than "Results" section.

Author's response:

We appreciate the reviewer's suggestion. We have accordingly put this section under "Methods" in the revision.

Comments:

The figures in the main text should be further improved. The text in most of figures are not clear to see, such as Figures, 2,3,4, and 5.

Author's response:

We appreciate the reviewer's comment. We have improved the quality of the figures in our revision version.

Comments:

For the description of the iDNA-ABF, they used the uppercase ABCD to order each module in main text, but in figure 1 and the legend, the lowercase abcd... is used. They should keep them consistent.

Author's response:

We appreciate the reviewer's comment. We have kept them consistent in our revision version.

Comments:

For the description of the iDNA-ABF, they used the uppercase ABCD to order each module in main text, but in figure 1 and the legend, the lowercase abcd... is used. They should keep them consistent.

Author's response:

We appreciate the reviewer's comment. We have fixed this problem and carefully checked the consistency in the article.

Comments:

The English of this manuscript should be further polished.

Author's response:

We appreciate the reviewer's comment. We have checked and polished the manuscript. In our current revision, it is easier to understand our method and experiments.

Comments:

For result reproductivity, the hyper parameters should be given.

Author's response:

We appreciate the reviewer's comment. We have provided related hyper parameters of our models in the Additional file (Table S11). Moreover, the files regarding model hyper parameters can be also downloaded from our server: <https://server.wei-group.net/idnaabf/>.

Comments:

In supplementary materials, I didn't find the results they mentioned in main text.

Author's response:

We appreciate the reviewer's comment. We have accordingly checked the tables and figures in the supplementary material for correspondence with the main text.

Responses to Reviewer #3

Comments:

The authors focused on 4-methylcytosine (4mC), 5-hydroxymethylcytosine (5hmC), and N6-methyldeoxyadenosine (6mA) are three DNA methylation types. They did not make clear whether a) they constructed three differently trained models to prediction of each type or they three times repeated a binary classification method for each methylation type. b) they constructed 3-class classification method. I guess they adopt (a), but (a) has little novelty because many scientists published the binary classification methods of DNA methylations.

Author's response:

We appreciate the reviewer's comments. In the previous version, we trained the models separately for different methylation types, and each methylation type prediction is a binary classification task. Indeed, we have accordingly built a new multi-task learning framework that can not only do the binary methylation classification task, but also do the regression task to predict the methylation signals. To be simplified, the regression task is using methylation related histone modification signals for model construction. Given a DNA sequence, our model can

directly predict the signal value and see if the sequence is methylated or not. However, the regression task highly relies on histone modification data for model training, which is not always available for some specific species or cell lines, especially for the species datasets used in this study. Thus, we didn't include the regression model in our main text, but presented the details of our regression model in Additional file (Supplementary Methods). The results are presented in Additional file. Moreover, for convenience of implementing our method, we have built a webserver, in which we particularly integrated the regression model together with those binary predictive models. This could provide users more options for methylation prediction.

Comments:

The authors need to investigate latest state-of-the-art predictors and are required to show some advantages over the latest predictors. It is important to compare their method with some latest predictors including Li Z, Jiang H, Kong L et al. Deep6mA: A deep learning framework for exploring similar patterns in DNA N6-methyladenine sites across different species, PLOS Computational Biology 2021;17:e1008767. Pian C, Zhang G, Li F et al. MM-6mAPred: identifying DNA N6-methyladenine sites based on Markov model, Bioinformatics 2019;36:388-392. Sho Tsukiyama, et al. BERT6mA: Prediction of DNA N6-Methyladenine site using deep learning-based approaches. Briefings in Bioinformatics 23:1-15, 2022.

Author's response:

We appreciate the Reviewer's comments. We have accordingly added brief description of these methods in "Introduction", conducted comparative experiments with these methods on the benchmarking datasets, and rewritten the "Result" section in the revision. Figure R3 illustrates the performance of our model and the state-of-the-art methods. The details of the results can be found in Additional file (Table S1).

Figure R3. Performance comparison between iDNA-ABF and other existing methods. (A) and **(B)** represent the ACC and MCC values of our proposed iDNA-ABF and other existing methods including iDNA-ABT, iDNA-MS, BERT6mA, and Deep6mA on 17 benchmark independent datasets, respectively. **(C)** The ROC and PR curves of our proposed iDNA-ABF and other existing methods in *5hmC_H.sapiens*. **(D)** The ROC and PR curves of our proposed method and other existing methods in *4mC_C.equisetifolia*. **(E)** The ROC and PR curves of our proposed iDNA-ABF and other existing methods in *6mA_C.equisetifolia*. **(F)** The ROC and PR curves of our iDNA-ABF and other existing methods in *6mA_F.vesca*.

Comments:

In Fig. 7, the authors visualized the attention scores of *4mC_S.cerevisiae*, *5hmC_M.musculus*, and *6mA_A.thaliana*. In Fig. 8 they performed motif matching of *4mC_Tolypo cladium*, *5hmC_H.sapiens* and *6mA_C. equisetifolia/F.vesca*. We like to see the consistency between the attention analysis and STREAME, but they used different species, which makes it difficult to see their consistency. They should conduct the sequence pattern matching by using the same species between Fig 7 and Fig 8.

Author's response:

Thank the reviewer for pointing out this question. We have addressed the corresponding problem in our revision. We conducted the sequence pattern matching using the same species to keep it consistency. Figures R2 (E-G) have clearly shown the relationship between the sequence patterns extracted by our model and the motifs identified from conventional motif analysis tool (STREME). To further quantify the similarity degree between the motifs extracted from iDNA-ABF and that from STREME, we calculated the p-value using TOMTOM. We demonstrate that our learnt motifs are significantly conserved.

Figure R4. Interpretable analysis of multi-scale information processing. (A) The comparison of single scales including 3-mer, 4-mer, 5-mer, and 6-mer, respectively. **(B)** The comparison of multi-scale combinations. **(C)** The attention map to illustrate the information capturing at 3-mer scale on one randomly selected sequence. Two sub-figures visualize the change of information capturing before and after training, respectively. **(D)** The attention map to illustrate the information capturing at 6-mer scale. **(E) - (G)** Interpretable illustrations of the motifs learnt by our model in three species covering three methylation types, including *4mC_Tolypo cladium*, *5hmC_H.sapiens*, and *6mA_C.equisetifolia*, respectively. The left part

figure clearly shows which region the model is more focused on by using heatmap from 0-1. The closer the score is to 1, the darker the color and the more important the region considered by the model. The p-value was calculated using TOMTOM by comparing our iDNA-ABF learnt motifs with STREME motifs. The p-value in STREME was calculated by a one-sided binomial test. The motifs within the gray dashed anchor boxes were extracted for pair comparisons.

Comments:

Since the attention analysis of DNA methylation has already been carried, they need to briefly discuss their attention analysis based on other works. For example, Sho Tsukiyama, et al. BERT6mA: Prediction of DNA N6-Methyladenine site using deep learning-based approaches. Briefings in Bioinformatics 23:1-15, 2022.

Author's response:

According to the reviewer's comments, we have briefly described the BERT6mA in Introduction, and also compared the predictive performance between BERT6mA and our model (see Figure 1 in main text). Moreover, we have re-designed the analysis procedure in our revision, in which we provided a comprehensive interpretable analysis for the DNA methylation predictions, which can help us build the mapping from the discovery of important sequential determinants to the in-depth analysis of their biological functions. Our interpretable analysis has the following advantages. (1) Using attention heatmap, we analyze what information our model learnt (see Figure R4(C-D)). (2) Using attention mechanisms, our model can learn the key sequential regions, which are most important for the prediction, along the input sequences (see Figure R4(E-G)). (3) Strong transfer learning ability of our model enables us capture the specificity of different methylation sequential patterns and learn some new functional motifs (see Figure R5(A-C)). (4) Our model can adaptively learn and extract the conserved motifs that are reported to be highly associated with the methylation mechanisms (see Figure R5(D)). Moreover, other important findings derived from the interpretable analysis can be found in "Results" and "Discussion" sections in main text. Altogether, different from existing work, we provide a more comprehensive and systematic analysis for the DNA methylation prediction.

Figure R5. Transfer learning result and analysis of our model. (A) The ROC and PR curves of the baseline model and the transfer learning model on CHG dataset in HepG2 cell line. Note that the baseline model is trained with the CHG dataset while the transfer learning model is pre-trained with CpG dataset and fine-tuned on the CHG dataset. **(B)** The ROC and PR curves of the baseline model and the transfer learning model in CHH dataset in HepG2 cell line. Note that the baseline model is trained with the CHH dataset while the transfer learning model is pre-trained with the CpG dataset and fine-tuned on the CHH dataset. **(C)** The motifs learnt from three models, including baseline model, pre-trained model, and transfer learning model, respectively. **(D)** The searching results using our learnt motifs against the topEnriched.MM database.

Comments:

In Fig. 4, to demonstrate the effectiveness of adversarial training, the authors need to compute the metrics (SN, SP, ACC, AUC, MCC,....) scores on the training datasets. An independent test is necessary to demonstrate the superiority of the adversarial training.

Author's response:

We appreciate the Reviewer's comment. In Figure R6, the adversarial training results are obtained from independent datasets, instead of training datasets. We have revised the article and added the corresponding details. The detailed results regarding the comparisons of other metrics (SN, SP, AUC) can be found in the Additional file (Figure S4). The comparative results demonstrated that the adversarial training in our training process alleviates the impact of large-scale parameters particularly on some small datasets, and improves the generalization ability of our model across different species and methylation types.

Figure R6. The MCCs and ACCs of the models with and without adversarial training on 17 benchmark independent datasets, respectively; each point in the figure represents each dataset.

End of Response

Second round of review

Reviewer 1

All my concerns have been addressed.

Reviewer 2

The author answered the question correctly.

Reviewer 3

It is improved.